# G-Quadruplex Structures Colocalize with Transcription Factories and Nuclear Speckles Surrounded by Acetylated and Dimethylated Histones H3

**DOI:** 10.3390/ijms22041995

**Published:** 2021-02-17

**Authors:** Denisa Komůrková, Alena Svobodová Kovaříková, Eva Bártová

**Affiliations:** Institute of Biophysics of the Czech Academy of Sciences, Department of Molecular Cytology and Cytometry, Královopolská 135, 612 65 Brno, Czech Republic; komurkova@ibp.cz (D.K.); aluskakovarikova@centrum.cz (A.S.K.)

**Keywords:** G-quadruplex structure, epigenetics, nuclear bodies, transcription factories, nuclear speckles, DNA repair

## Abstract

G-quadruplexes (G4s) are four-stranded helical structures that regulate several nuclear processes, including gene expression and telomere maintenance. We observed that G4s are located in GC-rich (euchromatin) regions and outside the fibrillarin-positive compartment of nucleoli. Genomic regions around G4s were preferentially H3K9 acetylated and H3K9 dimethylated, but H3K9me3 rarely decorated G4 structures. We additionally observed the variability in the number of G4s in selected human and mouse cell lines. We found the highest number of G4s in human embryonic stem cells. We observed the highest degree of colocalization between G4s and transcription factories, positive on the phosphorylated form of RNA polymerase II (RNAP II). Similarly, a high colocalization rate was between G4s and nuclear speckles, enriched in pre-mRNA splicing factor SC-35. PML bodies, the replication protein SMD1, and Cajal bodies colocalized with G4s to a lesser extent. Thus, G4 structures seem to appear mainly in nuclear compartments transcribed via RNAP II, and pre-mRNA is spliced via the SC-35 protein. However, α-amanitin, an inhibitor of RNAP II, did not affect colocalization between G4s and transcription factories as well as G4s and SC-35-positive domains. In addition, irradiation by γ-rays did not change a mutual link between G4s and DNA repair proteins (G4s/γH2AX, G4s/53BP1, and G4s/MDC1), accumulated into DNA damage foci. Described characteristics of G4s seem to be the manifestation of pronounced G4s stability that is likely maintained not only via a high-order organization of these structures but also by a specific histone signature, including H3K9me2, responsible for chromatin compaction.

## 1. Introduction

Information about the self-association of guanines to the quadruplex formation has been known since the 1960s [1]. Later, Sen and Gilbert in 1988 described the biological significance of G-quadruplexes in meiosis [2]. It is well-known that the basic structural motif of G-quadruplexes represents the Hoogsteen hydrogen-bonded guanine (G)-tetrad (also termed a G-quartet) [3], and it is believed that quadruplex sequences are distributed within the human genome in distinct regions, including exons, introns, untranslated regions, or promoter sequences. Much information is provided for telomere sequences and their ability to form G-quadruplex structures [4,5]. Telomere protection is believed to be maintained not only by shelterin proteins [6] but also via the function of both DNA and RNA G-quadruplexes [7]. By the use of a specific antibody, Biffi G. et al. [8] showed a subset of nuclear G4 structures located at telomeres. Moreover, Moye et al. [4] documented that telomerase colocalizes with G-quadruplexes in vivo. Therefore, it is likely that there is a structural and functional link between quadruplex structures and hTR, the RNA template of telomerase.

Fundamental is also the observation that that quadruplex sequences occur in the promoter regions of several cancer-related genes, including the c-Myc protooncogene [9,10]. Brooks and Hurley stated an important question, asking how G4s can be formed from stable B-DNA structure in a GC-rich region, such as the c-Myc gene [11]. Interestingly, these G-quadruplex structures can also be detected in genomic regions with chromosomal translocations [3]. Importantly, the c-Myc gene is also a locus involved in reciprocal translocations. For instance, the c-Myc/IgH translocation t(8;14) in Burkitt’s lymphoma [12]. Another example of translocation, accompanied by the formation of G-quadruplexes, represents t(14;18) that involves IgH and Bcl-2 genes. This structural cytogenetic aberration is associated with B-cell non-Hodgkin’s lymphoma [13,14]. From the view of genome instability, also genome-wide analysis in various cancer cells showed a hypomethylation and an enrichment in G4 structures close to DNA breakpoints, involving clusters of copy number alterations in the genome [15]. Moreover, Mao et al. showed that the DNA methyltransferase 1 (DNMT1) occupies G4-sites in the genome with the highest affinity rather than duplex- or the single-stranded DNA [16]. Therefore, it is likely that pathological hypomethylation of DNA and changes in G4 structures are associated with an aberrant karyotype leading to malignant cell transformation.

It is also interesting that G-quadruplex structures are formed in approximately 50% of human gene promoters; thus, G4s play a role in the regulation of gene expression [3]. G4 structure in DNA was published not only in the case of the c-Myc protooncogene [10] but also for the pluripotency gene Oct4 [17]. Moreover, it is well-known that G-quadruplex formation stabilizes not only the transcription of these genes but also regulates DNA replication and DNA damage response [18]. It is well-known that DNA replication is affected by the function of nucleases EXO1 and FEN1. From this view, Liu and Gilbert, along with Vallur et al., showed that these enzymes have the ability to cleave G-quadruplexes in vitro [19,20]. Interestingly, deficiency of these nucleases additionally induces telomere dysfunction [21].

From the view of DNA repair processes, Rodriguez et al. showed that a well-known DNA damage marker, phosphorylation of histone H2AX (γH2AX), is highly abundant in G4-enriched genes [22]. Thus, a question remains: How vital is the role of G-quadruplex structures in DNA damage repair or vice versa? From this view, it seems to be essential to understand which DNA repair proteins recognize G4 structures in the case of genome injury by DNA damaging agents.

Here, we come out from the statement that nuclear localization of G-quadruplex structures is not random and that G4 forms functionally essential genomic regions, like promoters or telomeres. Interestingly, a nuclear protein nucleolin interacts with G4s [23], and inhibition of G4-binding molecule, NM23-H2, caused a redistribution of nucleolin from the nucleolus to the nucleoplasm [11]. These data inspired us to study a reciprocal link among G4 structures and specific nuclear compartments, including nucleoli, SC-35 positive splicing speckles, Cajal bodies (CBs), PCNA-positive replication factories, RNAP II-positive transcription factories, promyelocytic leukemia (PML) nuclear bodies, or DNA damage foci. In addition, we studied an epigenetic landscape of G4s concerning histone signature. After that, we addressed how γ-radiation affects the number of G4 structures and to what extent the repair of G4s is mediated via phosphorylation of histone H2AX (γH2AX) or other nonhistone DNA repair proteins 53BP1 and MDC1. We tried to know more about the nuclear arrangement of G4s in distinct cell types, including human ESCs and tumor cells. We additionally addressed a question of how specific is the nuclear arrangement of G4 structures in individual cell cycle phases, especially the early and the late S-phase, because Biffi et al. showed on a synchronized cell population that the lowest degree of G4 formation could be observed in the G0/G1 phase of the cell cycle [8]. However, when the cells enter the replication S-phase, the number of G4s-positive foci significantly increases due to genome duplication. 

## 2. Material and Methods 

### 2.1. Cell Cultivation and Treatment

For experiments, we used human cervical carcinoma HeLa cells, HaCaT keratinocytes, and mouse embryonic fibroblasts (MEFs) or human embryonic stem cells (hESCs). Human tumor cells were cultivated in D-MEM (Dulbecco’s modified Eagle’s medium) supplemented with 10% fetal calf serum (FCS) (Merck, Prague, Czech Republic). Cultivation medium consisted of the following components: 1 μL of β-mercaptoethanol (#31350-010, Thermo Fisher Scientific, Brno, Czech Republic), 5 mL of nonessential amino acids (100×; #1140-035, Thermo Fisher Scientific), 5 mL of sodium pyruvate (#11360-039, Thermo Fisher Scientific), and 1.5 g of NaHCO_3_ (Lachema, Brno, Czech Republic). Human ESCs were cultivated following Bártová et al. [24].

For confirmation of the specificity of the antibodies against quadruplexes, we used treatments by DNAse and RNAse. After fixation, the cells were incubated with Turbo-DNAse (#AM2238, Invitrogene, Waltham, MA USA) or RNase (#R5503, Sigma-Aldrich, Prague, Czech Republic) for 30 min at 37 °C. The result of treatments is documented in Figure 1A,B showing that DNAse treatment reduced DAPI signal, and G4 structures were not detected. After RNAse treatment, G4s-positive foci were detected by the use of Anti-DNA G-quadruplex (G4) Antibody, clone 1H6 (MABE1126, Merck-Millipore, Prague, Czech Republic).

In addition, we tested the effect of γ-irradiation on the distribution properties of G4 structures. For irradiation by γ-rays, cells were cultivated on Petri dishes and irradiated by a dose of 5Gy of γ-rays delivered by cobalt-60 (Chirana, Prague, Czech Republic). Cells were fixed by methanol (MeOH) (−20 °C) for 15 min. Fixation was performed 2 h after γ-irradiation, and after that, immunofluorescence staining was done.

To study the effect of α-amanitin (#A2263, Merck, Germany), the RNA pol II inhibitor was used; the final concentration was 2 μg/mL; treatment was for 2 h.

### 2.2. Cell Transfection

For cell transfection, we used the METAFECTANE PRO system (Biontex Laboratories, GmbH, Planegg, Germany). The following plasmid constructs were used: RFP-tagged PCNA (a general gift from Prof. Cristina Cardoso, Darmstadt, Germany), and GFP-tagged hTRF-1 (a gift from C.M. Counter, Duke University Medical Center, Durham, NC, USA).

### 2.3. Immunofluorescence Staining

Immunofluorescence was modified following Bártová et al. (2011) or Stixová et al. (2012) [24,25]. For the detection of G-quadruplex structures, we optimized the immunofluorescence procedure by following way. The cells were fixed according to two distinct protocols. The first tested experimental approach was fixation in 4% formaldehyde (PFA) for 5 min at room temperature (RT); then 100 µL of 1% sodium dodecyl sulfate (SDS) was added for the next 5 min, and cells were permeabilized by 0.5% Triton X100 with Tween. The second way was the fixation in cooled (−20 °C) methanol (MeOH) for 15 min at RT without additional permeabilization. Then, 5% goat serum (blocking buffer), dissolved in PBS, was added for one hour at RT. Primary antibodies were diluted on blocking buffer and incubated with cells overnight at 4 °C. The next day, the cells were washed three times with 1× PBS and incubated with secondary antibodies for 1 h at RT. The following primary antibodies for the detection of G4s were used: Anti-DNA G-quadruplex structures, clone BG4 (MEBE917 Merck–Millipore, Czech Republic); and Anti-DNA G-quadruplex (G4) Antibody, clone 1H6 (MABE1126, Merck-Millipore, Prague, Czech Republic). In the case of anti-BG4, we used anti-FLAG, DYKDDDK Tag (D6W5B) Rabbit mAb (#14793, Cell Signaling, Danvers, MA USA). Incubation was 3 h at RT. To study a mutual link between G4s and DNA repair proteins, we used additional primary antibodies: antiphosphorylated histone H2AX (γH2AX; phospho S139, #ab2893, Abcam, Cambridge, UK), anti-53BP1 (#ab21083, Abcam, UK), anti-MDC1 (#ab11169, Abcam, UK), anti-SMD1 (#LS-C346290, LSBio, Seattle WA USA). A spatial link of G-quadruplex structures to specific nuclear compartments was studied by the use of an antibody against PML (#A1184, ABclonal, Abclonal, Woburn, MA, USA), nucleoli were visualized by antifibrillarin (#A1136, ABclonal, Abclonal, Woburn, MA, USA), and Cajal bodies were shown by the antibody against the coilin protein (#A6428, ABclonal, Abclonal, Woburn, MA, USA). Nuclear speckles were recognized by the use of an antibody against the splicing factor, SC35 (syn. SFRS2, #A12625, ABclonal, Abclonal, Woburn, MA, USA). In addition, we used anti-H4ac (#06-866, Merc Millipore), anti-H3K9ac (#06-942, Merc Millipore), anti-H3K9me2 (#07-441 Merc Millipore), and anti-HK9me3 (#ab8898, Abcam, UK). The following secondary antibodies were used: Alexa 488-conjugated goat antirabbit (#ab10077, Abcam, UK), and Alexa 594-conjugated goat antimouse (#A11032, Thermo Fisher Scientific, USA). Samples incubated without primary antibodies were used as a negative control staining. A contour of cell nuclei we visualized by 4′,6-diamidino-2-phenylindole (DAPI; Merck, Germany) dissolved in Vectashield (Vector Laboratories, Burligame, CA USA).

Note: clone 1H6 binds both tetramolecular and unimolecular (intramolecular) G-quadruplex (G4) structures in DNA, without sequence specificity, while exhibiting significantly lower affinity toward tetramolecular RNA structure or a triplex DNA structure and little or no affinity toward either non-G4 ssDNA or dsDNA. However, clone 1H6 does not seem to bind the [AGGG(TTAGGG)3] unimolecular G4 structures, indicating a broad selectivity toward many, but not all, DNA G4s [26].

### 2.4. Laser Scanning Confocal Microscopy

For image acquisition, we used a Leica SP-5 confocal microscope (Leica, Mannheim, Germany), equipped with a 63× oil objective (HCX PL APO, lambda blue) with a numerical aperture (N.A.) = 1.4. For image acquisition and analysis, we used the white light laser (WLL; wavelengths of 470–670 nm in 1 nm increments). Image acquisition was performed at a resolution of 1024 × 1024 pixels at 400 Hz frequency, and we used a bidirectional mode of scanning at 64 lines with a zoom 8×. For the data analysis, we monitored 30–40 cell nuclei, and performed an analysis of three biological replicates. We acquired several collections of three-dimensional (3D) confocal images with a resolution of 1024 × 1024 pixels at 400 Hz frequency. The number of foci, volume of foci, and size of nucleus, the colocalization coefficients (the Pearson’s correlation coefficient or colocalization rate) were calculated by using Leica LASX software. Deconvolution (Lightning, Leica, Mannheim, Germany) procedure was applied for the final versions of images. 

### 2.5. Statistical Analyses and Quantification of Fluorescence Intensity

Graphs were constructed in Excel software or by the use https://plotly.com/chart-studio/ (accessed date was between May–October 2020). The number of G4s-positive foci was calculated by the use of Leica LASX software, and we used its colocalization tool. For each experimental event, we analyzed 30–40 cell nuclei, and experiments were performed in triplicates. Data in all figures are shown as average value ± standard errors (S.E.) or as median plus minimal and maximal values (box plots). The Pearson’s correlation coefficient was also calculated using Leica LASX software. Data were analyzed by the Mann–Whitney U test (STATISTICA software), or the Student’s *t*-test (Sigma Plot 14.5 software, Systat Software, Inc., Palo Alto, CA, USA) was applied for statistical analyses.

## 3. Results

### 3.1. Immunodetection of G-Quadruplex Structures in DNA

The aim of this work was primarily to optimize the method of immunodetection of DNA G-quadruplex structures inside the cell nuclei. We tested two different antibodies and two approaches of cell fixation. The use of the BG4 antibody, in comparison to the 1H6 antibody, was published in the majority of research papers [8,16,27,28,29,30,31]. Unfortunately, the main disadvantage of this antibody is limited commercialization, and when the antibody is produced under laboratory conditions, there is limited stability of this compound. Therefore, for our experiments, we instead chose the 1H6 antibody recognizing G4 structures in DNA. The main advantage of this immunostaining is reduced detection time due to fast staining by primary and secondary antibodies without using of Flag-antibody for BG4. In this case, fluorescence signals were detected mainly inside the cell nucleus, as visualized by 3D-projection of confocal microscopy (Figure 1 and Figure 2A,B). The use of 1H6 antibody for the detection of G4s was, for instance, reported by many authors [26,28,32,33]. Importantly, Kazamier et al. drew attention to the possibility of the cross-reacting nonspecific 1H6 antibody with thymines found in single-stranded DNA [34]. In the case of BG4, DNA microarray experiments of thousands of G4s and non-G4 forming sequences show the antibody binding to G4s. In addition, a certain subset of single-strand DNA was recognized by this antibody despite the fact that these sequences do not form G4s [35]. The studies mentioned imply a limitation of the use of antibodies against G4s; thus, as every method, the use of anti-G4s also has its limit.

Due to the fact that we studied DNA quadruplexes and eliminated the detection of RNA quadruplexes, we tested the specificity of the antibody by treating the samples with DNAse and RNAse (see Section 2, Material and Methods, Figure 1A,B). We found that G-quadruplexes occur directly in DNA; thus, we tested the appearance of these structures in several compartments of the cell nucleus, including the nucleoli, chromocenters (clusters of centromeric heterochromatin; AT-rich sequences), replication foci, RNA Pol II-positive transcription factories, SC-35 positive nuclear speckles, PML bodies, Cajal bodies, and DNA repair foci.

### 3.2. Nuclear Localization of G-Quadruplex Structures Is in GC-Rich Chromatin and Outside the Fibrillarin-Positive Region of Nucleoli

We studied the localization of G-quadruplex structures in human cervical carcinoma cells HeLa. We observed that G4s are located outside DAPI-dense regions showing clusters of centromeric heterochromatin, called chromocenters. DAPI (4′,6-diamidino-2-phenylindole) binds to AT-rich sequences; thus, we can conclude that G4s are located in GC-rich genomic regions (Figure 2Aa,b). We also analyzed a link of G4s to the compartment of nucleoli, and we found an absence of G4s inside the fibrillarin-positive region of nucleoli (Figure 2B).

### 3.3. Epigenetic Landscape of G-Quadruplex Structures

It is well-known that histone post-translational modifications are specific regulators of gene expression. Proactivation markers are, for instance, histone acetylations and H3K36 trimethylation, while constitutive heterochromatin is characterized by a high density of H3K9 trimethylation. H3K9me2 has a rather silencing effect and stabilizes chromatin (Dong et al., 2020). Here, we studied both proactivating and silencing markers of gene expression. In this case, immunofluorescence analyses, combined with confocal microscopy, showed a high density of H4 acetylation in proximity to G4s (the Pearson’s correlation coefficient was 0.27 ± 0.1). Moreover, a very high degree of colocalization was between H3K9 acetylation and G4s structures (the Pearson’s correlation coefficient was 0.42 ± 0.07). Chromatin around G4s was also H3K9 dimethylated (the Pearson’s correlation coefficient was 0.44 ± 0.08), while colocalization between H3K9me3 and G4s was rare (the Pearson’s correlation coefficient was 0.15 ± 0.06) (Figure 3A–F).

### 3.4. Nuclear Distribution Pattern of G-Quadruplexes in Distinct Cell Types

We observed that the number of G-quadruplexes is different in distinct cell types. We studied human keratinocytes (HaCaT), cervical carcinoma cells (HeLa), human embryonic stem cells (hESCs), and mouse embryonal fibroblasts (MEFs). In these cells, we calculated the number of G-structures per nuclear volume, and we observed the highest number of G-quadruplexes per cell nucleus (volume), especially in human ES cells (Figure 4A,B). In these cells, we found the highest degree of variably in the number of G-quadruplexes and G4s foci were more robust in comparison to the other cell types studied (see median plus minimal and maximal values in Figure 4B).

### 3.5. The Link of G-Quadruplex Structures to Nucleoli, PML Bodies, Cajal Bodies, Transcription Factories, Nuclear Speckles, and Replication Foci: A Degree of Colocalization

For several nuclear structures, including nucleoli, promyelocytic leukemia (PML) bodies, Cajal bodies, nuclear speckles, transcription factories, and replication foci, we studied from the view of their spatial relationship to G-quadruplex structures in interphase nuclei (Figure 5A). As shown by the Pearson’s correlation coefficient, a very low level of colocalization was between G4s and fibrillarin (Figure 5A,B; Pearson’s correlation coefficient = 0.23 ± 0.05; colocalization rate = 16 ± 12%). Similarly, coilin-(Cajal bodies) and SMD1-positive foci partially colocalized with G4 structures (Figure 5A,B; the Pearson’s correlation coefficient for coilin was 0.23 ± 0.09, the Pearson’s correlation coefficient for SMD1 = 0.37 ± 0.09; colocalization rate for G4s and coilin = 16 ± 9%, and for G4s and SMD1 = 44 ± 0.1%). Comparison between G-quadruplex structures and PML bodies showed the Pearson’s correlation coefficient 0.43 ± 0.06, colocalization rate 43 ± 5%. G4s pronouncedly colocalized with SC-35 positive nuclear speckles (the Pearson’s correlation coefficient was 0.49 ± 0.04, colocalization rate 40 ± 6%). The most significant colocalization we observed between G4s and phosphorylated form of the RNA polymerase II (RNAP II) (the Pearson’s correlation coefficient was 0.68 ± 0.03; colocalization rate 60 ± 13%). Concerning PCNA-positive foci (replication factories), we observed that in the non-S phase of the cell cycle, the Pearson’s correlation coefficient was 0.35 ± 0.07 (colocalization rate was 32 ± 13). In contrast, in the late S-phase of the cell cycle (in PCNA-positive late replication sites), this coefficient was significantly lower; it was 0.15 ± 0.05 (Figure 5C,D). In the early S-phase, the Pearson’s correlation coefficient was 0.32 ± 0.05 (colocalization rate was 55 ± 12). This observation confirms the phenomenon mentioned above, that G4s, recognized by the 1H6 antibody, are localized in the proximity of early replicating chromatin, euchromatin (Figure 5D).

In detail, we also studied a link between G4s and so-called “transcription factories” and nuclear speckles, positive on the splicing factor, the SC-35 protein. We tested the phosphorylated form of RNAP II (RNAP II-phS5) that occupies specific regions in the cell nucleus (transcription factories). It is believed that GC-rich sequences are transcribed in these nuclear compartments. Thus, we tested if an inhibitor of RNAP II, α-amanitin, can affect the degree of colocalization between G4 structures and RNAP II foci. Similarly, we studied the effect of α-amanitin on G4s colocalization with SC-35 positive foci. In these cases, α-amanitin did not change the colocalization rate between G4s and given nuclear compartments that we learned (Figure 6).

### 3.6. G-Quadruplex Structures and DNA Repair

The γH2AX-positivity, 53BP1-(regulating NHEJ repair), and MDC1-positivity (regulating HR repair) in G4 structures are shown in Table 1; the highest colocalization rate is for G4s and γH2AX. However, in all cases studied, for HeLa and HaCaT cells, the Pearson’s correlation coefficient was only 0.2–0.29, which is the manifestation of a low degree of colocalization observed in nonirradiated and γ-irradiated cells (Figure 7A–C). Taken together, irradiation by γ-rays did not affect colocalization between G4s structures and DNA repair foci (Figure 7B,C; Table 1). Here, we also calculated the number of G4s before and after γ-irradiation, and our analysis showed a decrease in the number of G4s in γ-irradiated HeLa cells, but the Mann–Whitney U test did not confirm a statistically significant reduction of G4s foci caused by γ-irradiation (Figure 7D).

## 4. Discussion

G4 quadruplexes are four-stranded secondary structures of both DNA and RNA. It is well-known that these structures do not fulfill the condition of Watson–Crick base pairing; thus, the existence and exact functioning of G4s are discussed. It is believed that G4s play a role in genome integrity and regulate gene expression [36]. G-quadruplex RNA was found to be functional during several biological events, including pre-mRNA processing and translation [37]. It was published that G-quadruplex DNA structures regulate transcription, telomere elongation, and replication [38,39]. However, Bifi et al. showed a subtle colocalization rate between telomeric protein (shelterin protein) TRF1 and G4s [8]. We confirmed this phenomenon in our analysis, showing a very low colocalization rate, 3.58 ± 0.07%, between G4s and TRF1 (the Pearson’s correlation coefficient, calculated from 80 confocal sections, was 0.15 ± 0.05) (Figure 8). This observation is in agreement with Bifi et al. declaring 74% of BG4-positive foci outside telomeric sequences, abundant on the shelterin protein TRF2 [8]. From the view of epigenetic regulation, G4s appear at telomeres that are known to be H3K9 trimethylated and H4K20 trimethylated. This structural–epigenetic state stabilizes telomeric heterochromatin and controls the length of telomere sequences [40,41]. In this case, Polycomb group proteins also represent regulatory units. For example, PRC2 binds to TERRA and catalyzes H3K27 trimethylation. However, G-quadruplex RNA within pre-mRNA removes the PRC2 complex from a subset of genes; therefore, H3K27me3 is depleted [42]. From the view of epigenetic processes, we observed, due to the colocalization rate (31 ± 8%), that G4 structures are mainly H3K9 acetylated and H3K9 dimethylated, as it was also shown by colocalization rate (38 ± 10%) (Figure 3A–F). This observation fits well with the claim that G4s also appear in promoters of protein-coding genes, including PVT1, STARD8, or c-Myc, which chromatin in an active form is acetylated [17,43,44,45,46]. This is in good agreement with our observation showing an apparent colocalization of G4s with a transcriptionally active form of chromatin that appears to be positive on histone acetylation, RNAP II, and the SC-35 protein (Figure 3A,B and Figure 5A,B).

According to our data, it seems likely that there is specific compartmentalization of G4s in transcription factories and nuclear speckles (Figure 5A,B), and G4s decorate telomeric sequences. Thus, an appearance of G4s (visualized by the antibody against G4s [clone 1H6]) is mostly of nontelomeric origin (Figure 8A). However, immunofluorescence-FISH with G-quadruplex antibody BG4 and telomere-specific DNA probe showed the important biological role of G4s in telomeres in the middle S-phase of the cell cycle [4,47]. Interestingly, in vivo dynamics of telomeric G-quadruplex structures was published by Bryan [47]. Among nuclear structures, G4s were also found to be located inside the nucleolin-positive region. It is well-known that this multifunctional protein, nucleolin, can interact with nontraditional forms of RNA and DNA [11]. However, the binding of nucleolin to G4s may be very specific because nucleolin likely distinguishes between two versions of G-quadruplex loops [48]. G4s can form different 3D structures that allow binding other specific targets [11,49]. For instance, it was found that nucleolin may induce G4 formation in the proximity of the c-Myc gene promoter, and so it can regulate promoter activity of the c-Myc gene [50,51]. Nucleolin is also a marker of the nucleolus, so we tried to find a colocalization between G4 and the fibrillarin-positive region of the nucleolus. In this case, we did not observe direct colocalization between G4s and fibrillarin (Figure 2B).

Here, we also study the DNA damage process because chromatin immunoprecipitation (ChIP-Seq) determining the distribution of phosphorylation of histone H2AX (γH2AX) in the genome implied the existence of this DNA damage marker in computationally predicted G-quadruplex motifs [36]. Here, we observe that in nonirradiated cells, G4s are occupied by DNA damage markers, γH2AX, MDC1, and 53BP1, from ~30% (colocalization rate was 31–34% ± 9–10%; see Table 1). Irradiation by γ-rays seemingly reduced the degree of colocalization (see, colocalization rate 23 ± 7–11% in Table 1); however, distribution (e.g., maximal and minimal values of Pearson’s correlation coefficient) was relatively wide; thus, this parameter is not possible to consider as statistically significant. In all cases studied, the Pearson’s correlation coefficient was not changed when we compared nonirradiated and irradiated cells. The effect of irradiation did not also appear in the number of G4s (Figure 7D). These results imply that G4 structures are relatively stable in the genome exposed to γ-rays (Table 1, Figure 7C,D). This observation fits well with the data published by Spotheim-Maurizot and Davídková, or Kumari et al., showing that guanine in G4 structures has a radioprotective function [52,53]. This property of G4s that we observed in euchromatin (Figure 2A and Figure 5A,B) is despite the fact that more relaxed GC-rich chromatin (euchromatin) should be more sensitive to damage. However, for instance, Williamson et al. claimed that epigenetic modifications in chromatin affect its sensitivity to damage [54]. Jakob et al. additionally showed a local heterochromatin relaxation after damage and its relocation to the periphery of heterochromatin clusters (chromocenters) [55]. Interestingly, genome-wide mapping documented that lamina-associated heterochromatin at the nuclear periphery was more vulnerable to UV damage than euchromatin [40,56,57], so certain exceptions also exist in the vulnerability of euchromatin and especially heterochromatin to damage. Here, the Pearson’s correlation coefficient for G4s and DNA damage markers was relatively low (~0.2), and irradiation by γ-rays did not change the degree of colocalization between G4 and DNA repair foci (see Figure 7A–D). This fact led us to the conclusion that G4s have a radioprotective function that is likely ensured by histone signature, especially H3K9me2, which should be that regulatory marker because it is, in general, responsible for chromatin compaction and thus stability (Figure 3C,E,F). 

Together, G4s structures are formed in both telomeric and nontelomeric DNA sequences [8], Figure 2A, Figure 5A,B and Figure 8. Here, from 50–60%, G4s colocalize with transcription factories and nuclear speckles; thus, they appear in proximity to transcription sites that are likely spliced cotranscriptionally (Figure 5A,B). This observation is in agreement with Georgakopoulos-Soares et al. showing that an appearance of G4s in the sites of the alternative splicing could be one possibility of splicing regulation [58]. In general, these data document a high density of G4s in GC-rich (euchromatin) sequences. This claim is also confirmed by the fact that 31 ± 8% of G4s is H3K9 acetylated, which is an epigenetic marker of transcriptionally active and decondensed chromatin. From the view of DNA repair, we found that proteins of DNA damage response, to some extent, colocalize with G4s, but γ-irradiation does not affect the degree of this colocalization; thus, the stability of G4 structures is relatively high.

## Figures and Tables

**Figure 1 ijms-22-01995-f001:**
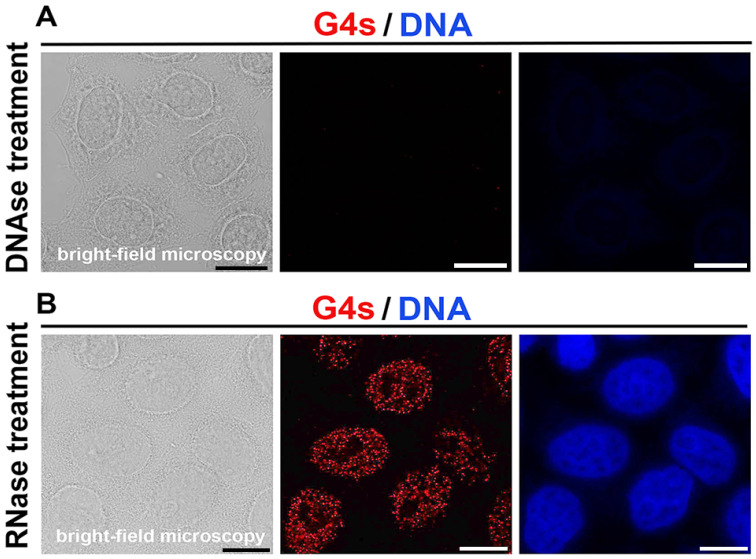
Optimization of G-quadruplexes (G4s) detection. Samples were treated by (**A**) Turbo-DNAse or (**B**) RNase for 30 min at 37 °C, and after immunofluorescence (Anti-DNA G-quadruplex (G4) Antibody, clone 1H6 was used), the presence/absence of G4s structures (red) was analyzed inside interphase nuclei (blue). In both panels, cells were additionally visualized in transmission light (the bright field microscopy). Scale bars indicate 20 µm.

**Figure 2 ijms-22-01995-f002:**
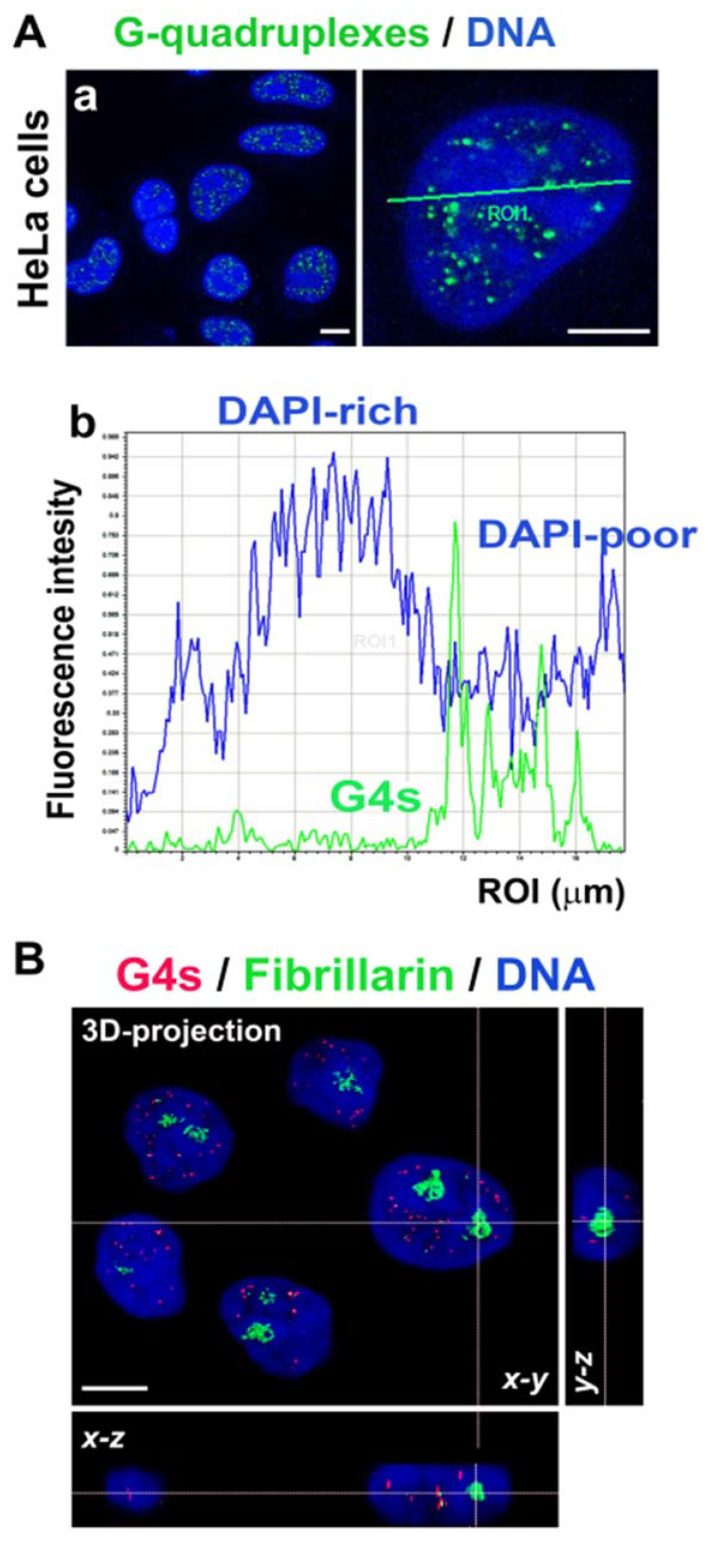
G4 structures (green) were located in (**Aa**) 4’,6-diamidino-2-phenylindole (DAPI)-poor chromatin (blue), as it was quantified according to (**Ab**) fluorescence intensity by the Leica LAS X software. (**B**) G4 structures (red) do not colocalize with the fibrillarin-positive region of nucleoli (green). DAPI (blue) staining was used for visualization of the cell nuclei that are shown in the 3D-projection of the confocal image. Scale bars show 10 μm.

**Figure 3 ijms-22-01995-f003:**
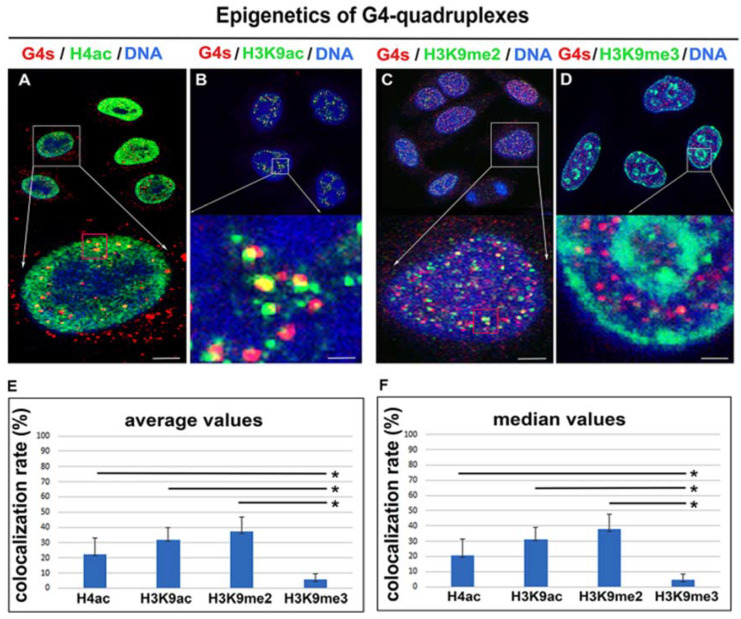
Histone signature of G4 structures in HeLa cells. (**A**) G4s (red) structures were associated with acetylated histones H4 (green). (**B**) G4s (red) and H3K9ac (green). (**C**) G4s (red) and H3K9me2 (green). (**D**) G4s (red) and H3K9me3 (green). DAPI (blue) was used as a counterstaining of the cell nucleus. Scale bars represent 3 µm in panels A, C and 1 μm in panels B, D. Arrows show enlarged selected cells or selected regions inside cell nuclei. Panels show the rate of colocalization between G4s and H4ac, G4s and H3K9ac, G4s and H3K9me2, G4s and H3K9me3. Panel (**E**) illustrates data documenting colocalization rate and its average values. Panel (**F**) shows the colocalization rate in terms of median values. Data are presented as a colocalization rate with standard errors (S.E.). The normality test (Shapiro–Wilk) passed for comparison of G4s/H4ac and G4s/H3K9me3. In this case, the difference in the mean values of the two groups was greater than would be expected by chance; there is a statistically significant difference between the input groups (*p* ≤ 0.001). The Student’s *t*-test revealed statistically significant differences, as indicated by asterisks (*). As reference values, the colocalization rates of G4s/H3K9me3 were used. The normality test (Shapiro–Wilk) failed for H3K9ac (H3K9me2) and H3K9me3, but the Mann–Whitney rank sum test showed that the difference in the median values between the two groups was greater than would be expected by chance; there was a statistically significant difference (*p* ≤ 0.001).

**Figure 4 ijms-22-01995-f004:**
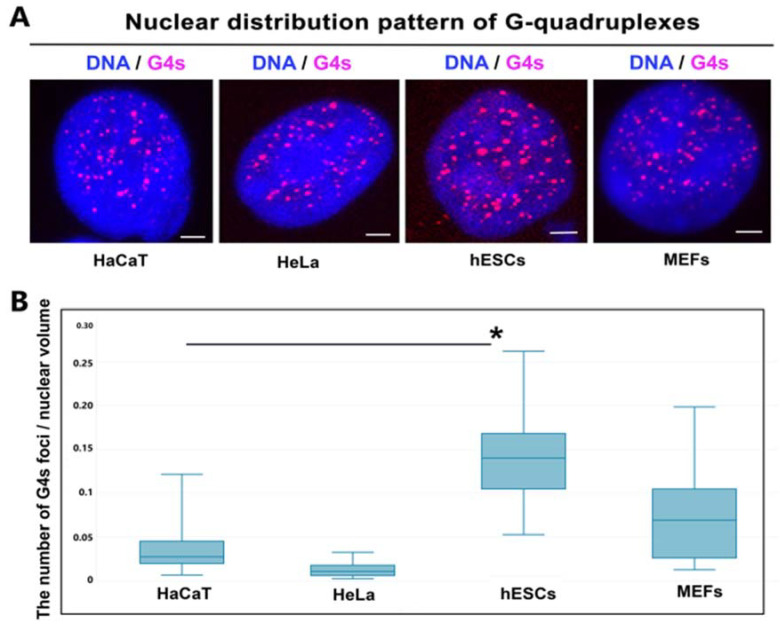
Nuclear distribution of G4s in distinct cell types. (**A**) Confocal microscopy of G4s in the following cell types: HaCaT, HeLa, hESCs, and MEFs. (**B**) Analysis of the number of G4s per nuclear volume of individual cells studied in panel (**A**). For analysis, we used 30–40 cell nuclei (experiments were repeated three times). Leica LAS X software was used for data analysis. Lines inside box plots show medians; 50% of the data is inside the box plots, and line segments show border values. Data are shown as the average number of G4s foci ± standard errors (S.E.). Bars show the lowest and highest value. When using the Student’s *t*-test, the normality test (Shapiro–Wilk) failed (*p* < 0.050), but the Mann–Whitney rank sum test revealed that the difference in the median values between the two groups (HaCaT and hESCs) is greater than would be expected by chance; there was a statistically significant difference (*p* ≤ 0.001). A statistically significant difference is indicated by an asterisk (*). As a reference value, the number of G4s in HaCaT cells was used.

**Figure 5 ijms-22-01995-f005:**
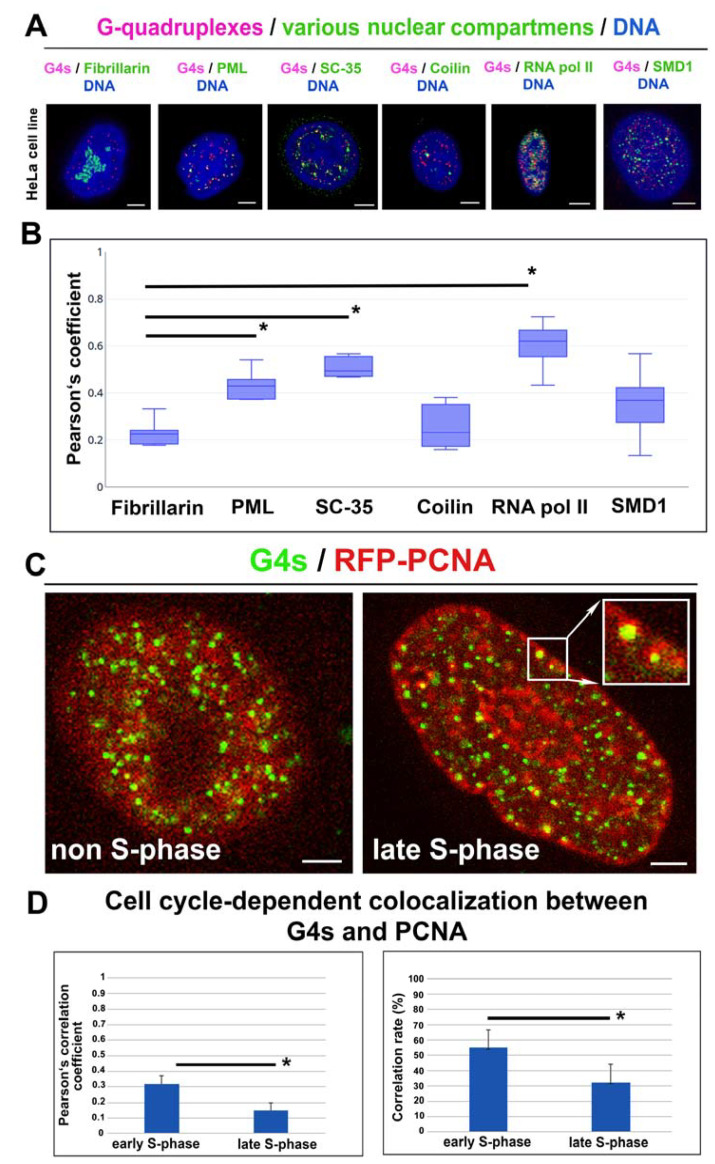
The spatial relationship between G4s and nucleoli, PML bodies, SC35-positive nuclear speckles, Cajal bodies, RNAP II-positive transcription factories, and PCNA-positive replication foci. (**A**) The nuclear distribution pattern of G4s (red) and fibrillarin (green), PML nuclear bodies (green), the SC35 protein (green), the coilin protein (green), phosphorylated form of RNAP II (green), and the SMD1 protein is shown. (**B**) Analysis of a degree of colocalization between G4s and nuclear compartments. In the cases indicated by an asterisk (*), the equal variance test (Brown–Forsythe) passed, similarly, the normality test (Shapiro–Wilk), so that the Student’s t-test showed the difference in the mean values of the two groups as greater than would be expected by chance; there is a statistically significant difference between the input groups (*p* ≤ 0.001). As a reference, the Pearson’s correlation coefficient was used for those observed for G4s and fibrillarin. Bars show the lowest and highest level of the Pearson’s correlation coefficient. Data in panel B represent the median of the Pearson’s correlation coefficient with minimal and maximal values. (**C**) The reciprocal link between G4s and PCNA-positive foci in non-S phase cells and the late S-phase of the cell cycle. Data originate from 30–40 nuclei analyzed for each experimental event. In panel (**D**), the Mann–Whitney U test was used for the statistical analysis, showing differences between G4s colocalization with PCNA-positive replication foci in the early S-phase and the late S-phase of the cell cycle. Asterisks indicate statistically significant differences at α = 0.05.

**Figure 6 ijms-22-01995-f006:**
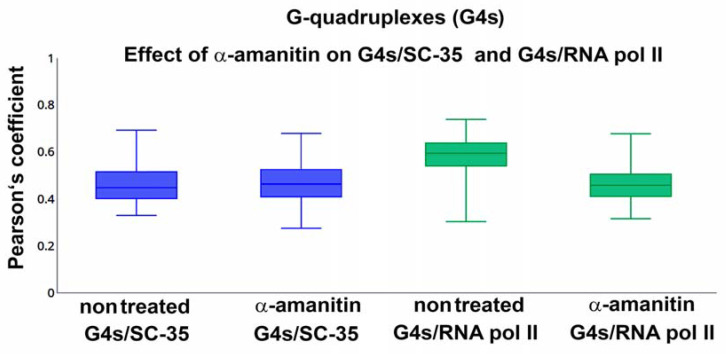
Colocalization between G4s and SC35-positive nuclear speckles or G4s and RNAP II-positive transcription factories was not affected by the inhibitor of RNAP II, α-amanitin. Data are shown as the median of the Pearson’s correlation coefficient with minimal and maximal values. Bars show the lowest and highest values of the Pearson’s correlation coefficient. The Mann–Whitney U test was used for the statistical analysis. The test revealed that differences between measurements are not statistically significant.

**Figure 7 ijms-22-01995-f007:**
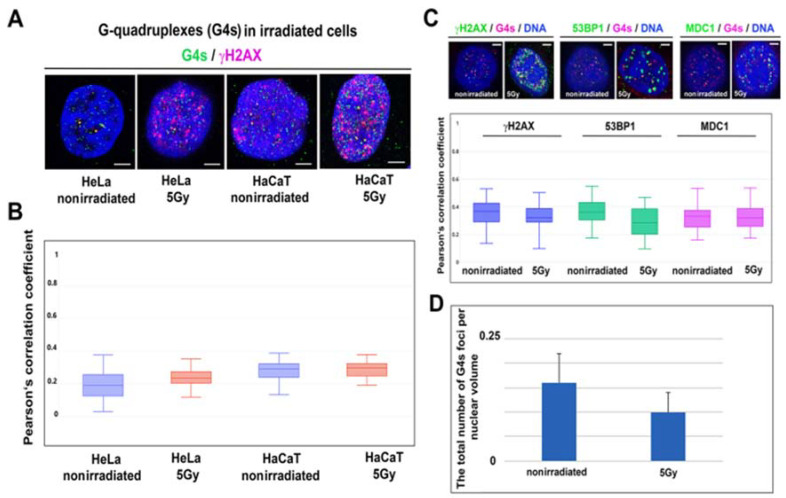
Irradiation by γ-rays does not change the γH2AX-, 53BP1-, MDC1-positivity in G4 structures, as shown in panels (**A**–**C**), in HeLa and HaCaT cells. Data in panels (**A**,**B**) originate from experiments performed by the use of BG4 antibody, and for data in panels (**C**,**D**), we used 1H6 antibody. Panel (**D**) shows the number of G4s/nuclear volume calculated in nonirradiated and γ-irradiated cells. Scale bars show 5 µm. The Mann–Whitney U test was used for the statistical analysis, but no significant differences were observed in G4s and DNA repair proteins studied in nonirradiated and γ-irradiated cells. The difference in the median values between the two groups (nonirradiated and irradiated cells) is not great enough to exclude the possibility that the difference is due to random sampling variability; there were no statistically significant differences (*p* = 1.000).

**Figure 8 ijms-22-01995-f008:**
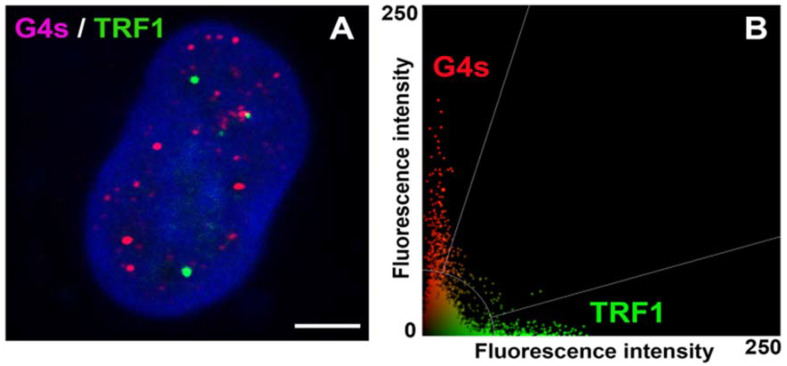
Example of the degree of colocalization between shelterin protein TRF1 (green), associated with telomeres, and G4 (red) structures in HeLa cells. Data were analyzed by Leica LAS X software and originate from one selected confocal section; thus, only several telomeres (green) are visible. (**A**) Confocal images show the distribution profile of GFP-tagged TRF1 (green) and G4s structures (red). Scale bar shows 5 μm. (**B**) The scatter plot (Leica LAS X software) shows a low degree of colocalization between the TRF1 protein and G4s.

**Table 1 ijms-22-01995-t001:** Degree of colocalization between G4 structures and DNA repair proteins.

Colocalization Partners- Treatment	Pearson’s Correlation Coefficient (PCC)	Correlation Rate (%)
G4s/γH2AX-control	0.37 ± 0.09	34 ± 9
G4s/γH2AX-5Gy	0.32 ± 0.08	23 ± 7
G4s/53BP1-control	0.36 ± 0.08	31 ± 10
G4s/53BP1-5Gy	0.28 ± 0.10	23 ± 11
G4s/MDC1-control	0.33 ± 0.08	27 ± 9
G4s/MDC1-5Gy	0.32 ± 0.08	38 ± 8

The analysis was performed by using Leica X software (Leica, Mannheim, Germany).

## Data Availability

Original micrographs (files in gigabytes, GB) are on-demand; please address Eva Bártová (e-mail: bartova@ibp.cz).

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
