# Peer review of "G-Quadruplex Structures Colocalize with Transcription Factories and Nuclear Speckles Surrounded by Acetylated and Dimethylated Histones H3"

_ijms, 2021, doi:10.3390/ijms22041995_

Round 1

Reviewer 1 Report

This manuscript describes sub-nuclear localization of G-quadruplex structures of DNA,   investigated by an immunostaining assay using anti G4 antibody, 1H6, together with a series of nuclear compartment markers in several mammalian cell lines. The reviewer has already reviewed the original manuscript of this study and finds one original critique to be unsatisfactory in this revised manuscript. The authors should address this point before publication.

  1. Throughout the manuscript, comparison of experimental data based on statistical significance is lacking. The authors should perform statistical analysis, such as a standard t-test or other appropriate statistical tests, for all statistical data to discuss the statistical significance of each event.

Author Response

The first reviewer

We thank the reviewer for his/her additional suggestions and comment regarding how to improve our manuscript. Herein, we have replied to the reviewers’ criticisms. Changes are shown in red fonts.

This manuscript describes sub-nuclear localization of G-quadruplex structures of DNA,   investigated by an immunostaining assay using anti G4 antibody, 1H6, together with a series of nuclear compartment markers in several mammalian cell lines. The reviewer has already reviewed the original manuscript of this study and finds one original critique to be unsatisfactory in this revised manuscript. The authors should address this point before publication.

  1. Throughout the manuscript, comparison of experimental data based on statistical significance is lacking. The authors should perform statistical analysis, such as a standard t-test or other appropriate statistical tests, for all statistical data to discuss the statistical significance of each event.

Answer: the Student’s t-test was used for statistical analysis in Fig. 3 E, E, Fig. 4B, and Fig. 5B. If data normality was not confirmed by Sigma Plot software, the Mann Whitney test was applied.

Reviewer 2 Report

Manuscript by Denisa et al. evaluates colocalization of G-quadruplexes with transcription factories and nuclear speckles surrounded by acetylated and dimethylated histones. Upon evaluation of variability in the number of G-quadruplexes in selected human and mouse cell lines the highest number of G-quadruplexes was found in mouse embryonic stem cells. A high colocalization was found between G-quadruplexes and other cell compartments and phosphorylated form of RNA polymerase II. Irradiation by γ-rays did not change a mutual link between G-quadruplexes and DNA repair proteins

Hypothesis and questions raised are timely. Technically study is performed at a very high level. Results and their discussions are well supported with recent literature citations.

Author Response

Answer: We thank the reviewer for his/her recommendation.

Round 2

Reviewer 1 Report

Accept in present form

This manuscript is a resubmission of an earlier submission. The following is a list of the peer review reports and author responses from that submission.

Round 1

Reviewer 1 Report

This manuscript describes sub-nuclear localization of G-quadruplex structures of DNA (G4DNAs) investigated by an immunostaining assay using anti G4 antibody, 1H6, together with a series of nuclear compartment markers in several mammalian cell lines. The authors demonstrate that the number of G4 foci detected by 1H6 in nuclei was varied on the dependence of cell types, and the G4 foci highly colocalize with H4ac, H3K9ac, and H3K9me2, which are seen in a relatively less compact chromatin, compared to H3K9me3, a heterochromatin marker. G4 foci of nuclei are found to be also enriched in the sits of transactional machineries and nuclear speckles compared to the other nuclear compartments, regardless of functional activities of these sites. The authors further show that the dynamicity of G4 foci is less observed during cell cycles and before/after the irradiation of γ-rays, in terms of the colocalization with a replication protein marker and a DNA damage marker, respectively. These findings highlight the spatial distribution of G4DNAs in nuclei and offer an insight into the potential function of G4DNAs in mammalian cells. However, several concerns should be addressed before the manuscript is acceptable for publication.

  1. The title should be reconsidered and amended. The word “H3K9/H4 acetylated and H3K9 dimethylated G-quadruplex structures” is confusing. It is histones, not G-quadruplex structures that are acetylated and dimethylated.

  1. A statement “These characteristics of G4s in protein-coding loci” seems not correct. Colocalization of RNAP II with G4 foci does not mean that G4s are located in protein-coding loci because of the limited resolution of a standard immunostaining assay as performed in this study. The author should consider rewording this sentence.

  1. The reviewer could not find any G4 immunostaining data of RNase and DNase-treated cells throughout the manuscript. The authors should provide this data.

  1. The reviewer is wondering how to know that 1H6 foci in nuclei are derived from G4 DNAs. G4s can be formed in pre-mRNAs or noncoding RNAs that locate at nuclei. If the authors want to investigate G4DNAs using immunostaining, all experiments must be treated with RNase to eliminate the nuclear RNA moleucles. Considering that the present study only relies on the immunostaining assay using 1H6, which is highly cross-active to poly thymine sequences as acknowledged by the authors, RNase treatment seems to be essential for avoiding the detection of G4 RNAs in nuclei and accurately investigating the sub-nuclear localization of G4DNAs in all experiments.

  1. The G4 immunostaining data obtained from cells exposed to the irradiation of γ-rays are questionable. As mentioned readily in this manuscript and ref. 28, guanine residues participating in G4 structures are indeed more protective for the γ-ray DNA damage than ones of non-G4-forming GC-rich sequences. However, it is known that G4-forming sequences are usually located in GC-rich sequences, which are greater sensitive to the γ-ray DNA damage. Given that guanines are the most oxidative among the other nucleobases, the increased G4 colocalization with DNA damage markers seems to be observed at the limited resolution of a standard immunostaining assay as performed in this study. The authors should discuss this issue.

  1. The authors should add an appropriate explanation for why there is no significant correlation between the PCNA protein and G4 structures in the late S-phase cells compared to that in no S-phase cells.

  1. There are no data and discussions of the number of G4 foci after/before the irradiation and between non-S phase and the late S-phase cells. This information is important for assessing G4 formation as well as the validation of the antibody. Appropriate data and discussions should be presented in the revised manuscript.

  1. Throughout the manuscript, a comparison of experimental data based on statistical significance is lacking. The authors should perform statistical analysis such as a standard t-test or other appropriate statistical tests to discuss the statistical significance of each event of experimental data.

Reviewer 2 Report

In this study Komůrková et al. investigated the nuclear localisation of G4 structures. They examined the localisation of G4 in specific nuclear compartments and the colocalization of G4 with specific histone marks and with DNA repair proteins. Although this is an interesting question, the design of this study is flawed in my view due to the fact that in another article Kazemier HG et al. showed that 1H6 cross reacts with other DNA structures and particularly with thymidines in single stranded DNA (Kazemier HG et al. NAR 2017). Kazemier HG et al. warned that observations of cross reactivity “complicate the interpretation of 1H6 binding to (sub-) cellular structures and need to be considered when this antibody is used.” Kazemier et al also stated: “Our confidence that G4 structures are also recognized by 1H6 in metaphase chromosomes and heterochromatin is considerably less in view of the novel observation reported here.” And they also concluded: “The new findings regarding the specificity of 1H6 reported in this study complicate the interpretation of 1H6 staining results and point to the need to perform additional studies to confirm that 1H6 binding to DNA indeed reflects the presence of G4 structures.” As a result, 1H6 antibody should not be used to investigate nuclear localisation of G4 structures until proven that this antibody actually reflects the presence of G4 structures. In that regard, I cannot accept this manuscript for publication in IJMS.